

# Strong correspondence between nitrogen isotope composition of foliage and chlorin across a rainfall gradient: Implications for paleo-reconstruction of the nitrogen cycle

Sara K. E. Goulden[1], Naohiko Ohkouchi[2], Katherine H. Freeman[3], Yoshito Chikaraishi[2], Nanako O. Ogawa[2], Hisami Suga[2], Oliver Chadwick[4], Benjamin Z. Houlton[1]

[1]University of California at Davis, Davis, CA 95616 USA
[2]Japan Agency for Marine Earth-Science and Technology (JAMSTEC) Yokosuka, Kanagawa 237-0061, Japan
[3]The Pennsylvania State University, University Park, State College, PA 16801 USA
[4]University of California Santa Barbara, Santa Barbara, CA 93106 USA

*Correspondence to:* Sara K.E. Goulden (skegoulden@gmail.com)

**Abstract**. Nitrogen (N) availability influences patterns of terrestrial productivity and global carbon cycling, imparting strong but poorly resolved feedbacks on Earth's climate system. Central questions concern the timescale of N cycle response to

elevated $CO_2$ concentration in the atmosphere, and whether availability of this limiting nutrient increases or decreases with climate change. Nitrogen isotopic composition of bulk plant leaves provides information on large-scale patterns of N availability in the modern environment. Here we examine the utility of chlorins, degradation products of chlorophylls, hypothesized to persist in soil subsequent to plant decay, as proxies for reconstructing past plant $\delta^{15}N$. Specifically, we test the hypothesis that $\delta^{15}N$ of plant leaves ($\delta^{15}N_{leaf}$) is recorded in $\delta^{15}N$ of pheophytin *a* ($\delta^{15}N_{pheo}$) along the leaf-litter-soil continuum

across an array of ecosystem climate conditions and plant functional types (C3, C4, legumes, and woody plants). The $\delta^{15}N$ of live foliage and bulk soil display marked declines with increasing rainfall, consistent with past studies in Hawaii and patterns worldwide. We find measurable chlorin concentrations along soil-depth profiles at all sites, with pheophytin *a* present in amounts required for isotopic analysis (>10 nmol). $\delta^{15}N_{pheo}$ in leaves, litter, and soil track $\delta^{15}N_{leaf}$ of plant leaves. We find potential for $\delta^{15}N_{pheo}$ records from soil to provide proxy information on $\delta^{15}N_{leaf}$.

## 1. Introduction

A combination of high biological requirements, limited natural supply, and high chemical reactivity and mobility gives nitrogen (N) important controls over photosynthesis and respiration in the terrestrial biosphere (Vitousek and Howarth, 1991). The response of the terrestrial carbon (C) cycle to climate change and increased atmospheric carbon dioxide ($pCO_2$) will

depend largely on the availability of N (Hungate et al., 2003;Wang and Houlton, 2009;Ainsworth and Long, 2005;Denman, 2007;Thornton et al., 2009), raising the question of how N availability may also be expected to change.





Humans have modified the N cycle directly through production of reactive N (Galloway et al., 1995) and likely indirectly through several climate-induced changes receiving ongoing investigation, including changes in biological demand (McLauchlan et al., 2013), changes in organic mineralization (Durán et al., 2016) and mineral weathering rates (Houlton et al.,

2018), and changes in N-fixation rates. Experimental manipulations have yielded tremendous insights into mechanisms underlying complex carbon-climate-N cycle dynamics, yet limitations of geography, complexity, and time have left open questions about the overall N biogeochemical response to climate change at the landscape and global scales.

Of particular concern is the hypothesis of progressive nitrogen limitation (PNL) under $CO_2$ enrichment. This set of ideas

predicts that under rising atmospheric $pCO_2$, feedbacks between plants, soil, and microbes result in diminishing quantities of available N, thereby constraining plant productivity over time. This hypothesis is supported by results from some annual-to-decadal-scale experiments, including free air $CO_2$ enrichment (FACE) studies ((Luo et al., 2004;Ainsworth and Long, 2005) and citations therein). Others have pointed to limitations in this model whereby plant-soil-microbe systems adjust to changes in C by N interactions in a way that allows for sustained plant $CO_2$ uptake (e.g. (Drake et al., 2011)). On decade to millennial

timescales, with time and space for species shifts and turnover of soil resources, and with environmental conditions such as temperature and moisture concurrently changing, major N gain and loss pathways can be expected to change under a warming world in ways that could prevent PNL. For example, $N_2$-fixation may increase, either due to lower energetic barriers or increases in populations of $N_2$-fixers in plant/microbial communities. In terms of N losses, denitrification rates may increase, either directly or through respiration-driven changes in soil oxygen availability. Changes in moisture that accompany the

changes in temperature may augment or counteract the effects of temperature on these pathways or on decomposition (Parton et al., 1996;Greaver et al., 2016).

Insight into actual outcomes on real landscapes over decadal-to-millennial timescales would be helpful to predicting how feedbacks between C and N cycles will play role in climate change this century (Luo et al., 2004). If we could observe how N

availability to plants responded to past periods of change in climate and $pCO_2$, such as during transitions from glacial to interglacial periods, we could improve our projections for future integrative responses of the N cycles to a complex, changing world. Knowledge of past N cycle behaviour would provide both insight into long-term dynamics and important baselines from which we can better assess anthropogenic impacts.

Natural abundance stable isotopes of N ($\delta^{15}$N) are useful integrators of N cycle processes in modern environmental systems (Robinson, 2001), and are natural sources of evidence for N cycle behaviour in the past. Due to high reactivity and mobility of N, point-based concentration measurements give temporally limited views of dynamic plant-soil-microbial cycling of N. $\delta^{15}$N of plants and soils reflect a time-integrated signal of the N cycle. The gaseous loss processes (denitrification, ammonia volatilization, and anammox) which fractionate the light and heavy isotopes in terrestrial ecosystems are substrate-dependent,




making $\delta^{15}N$ sensitive to changes in the availability of nitrogen (Houlton and Bai, 2009). Conversely, at low N availability, there is likely to be less isotopic expression of these pathways, as well as potential for greater dependence of plants on ectomycorrhizae, thereby reducing isotopic difference between plant foliage and N sources (Hobbie and Ouimette, 2009). As a result, higher $\delta^{15}N_{leaf}$ corresponds to higher N availability to plants on average. (e.g., (Handley et al., 1999;Amundson et al.,

2003;Houlton et al., 2007;Craine et al., 2009;Martinelli et al., 1999)). Reconstructions of $\delta^{15}N_{leaf}$ would accordingly provide information on availability of N to plants in the past.

The same reactivity of N that makes $\delta^{15}N_{leaf}$ a valuable proxy for N availability in modern landscapes makes obtaining a primary N isotope signal from dead, buried, and decomposed organic matter particularly challenging. Primary $\delta^{15}N$ signatures

are altered by decomposition and diagenesis (Thackeray, 1998;Meyers and Ishiwatari, 1993;Hedges and Oades, 1997), and bulk interpretations are confounded by preferential preservation and accumulation of macromolecules with distinct $\delta^{15}N$ values (Hobbie and Ouimette, 2009;Wedin et al., 1995). Because low-oxygen environments preserve organic matter, sediment accumulations in small lakes have been used as sources of terrestrial $\delta^{15}Nleaf$ (McLauchlan et al., 2013). However, lake ecosystem processes can obscure the original $\delta^{15}N$ of terrestrial sources (Meyers and Ishiwatari, 1993). Land-based paleo-

proxy $\delta^{15}N_{leaf}$ inferences have targeted bulk N protected structurally, in fossil faunal material (Stevens and Hedges, 2004), in tree wood (Hietz et al.), and, in one case, pollen (Descolas-Gros and Scholzel, 2007). These approaches require controversial case-by-case defence of the primary nature of $\delta^{15}N$ due to diagenetic vulnerability (Harbeck et al., 2004;Thackeray, 1998) and redistribution of N (Gerhart and McLauchlan, 2014). In the case of faunal material, it is also necessary to consider species-specific trophic enrichment factors, age-effects, and dietary protein content (Sponheimer et al., 2003;Overman and Parrish,

2001). Many of these approaches are further limited for understanding landscape-level patterns by the poor spatial distribution of samples.

Compound-specific isotope analysis (CSIA) of sedimentary material offers advantages over these methods: Isotope ratios of individual compounds can be more resistant to diagenesis than those of bulk materials, and have the further advantages of

deriving from more constrainable sources. This last characteristic permits analysis of material from integrative depositional environments. In marine N biogeochemistry, subaqueous sediments that collect and bury organic compounds in time series deposits are widely used for paleoenvironmental reconstruction (Junium et al., 2011;Meyers, 1997). Soils likewise accumulate organic material over time, though not perfectly analogously to subaqueous systems.

The ideal characteristic compound for CSIA reconstruction of $\delta^{15}N_{leaf}$ in soil is N-rich, produced only above-ground, and retains the primary $\delta^{15}N$ of plant leaves. We suggest that degradation products of the chlorophyll molecule (chlorins) meet all the above criteria. Chlorins derive from chlorophyll through processes of senescence, decomposition, diagenesis, and grazing (Keely, 2006;Treibs, 1936). They have been successfully extracted from plants, litter, organic soil layers, sediments, and coal



deposits (Kennicutt et al., 1992;Sanger, 1971;Hodgson et al., 1968;Bidigare et al., 1991;Dilcher et al., 1970), but have not previously been sought in mineral soil horizons.

$\delta^{15}N$ of chlorins derived from algal pigments deposited in sediments have been used to infer N cycling processes in aquatic

systems (Higgins et al., 2010;Sachs and Repeta, 1999;Enders et al., 2008;Tyler et al., 2010), as the biological offset in algae is relatively well-constrained (Higgins et al., 2011;Sachs et al., 1999). Several studies have shown that the $\delta^{15}N$ of plant chlorophyll retains a bulk foliar $\delta^{15}N$ signature with an even smaller offset (Chikaraishi et al., 2005;Bidigare et al., 1991;Kennicutt et al., 1992), but landscape effects such as climate on this offset have not been explored.

$\delta^{15}N$ has not previously been measured on chlorins extracted from decomposed plants, terrestrial organic matter, or soil, and the behavior of chlorophyll $a$ degradation products and their $\delta^{15}N$ values along the leaf-soil continuum is yet to be explored. Understanding the fate of pheo $a$ and $\delta^{15}N$ of pheo $a$ is essential; this will determine whether systematic or non-systematic effects confound its utility as a paleo-tool.

Here we examine $\delta^{15}N$ of pheo $a$ pools in leaves, litter, and soil and compare values to bulk foliar sources to evaluate this compound as a proxy for past terrestrial N cycling. For chlorins to be a useful foliar $\delta^{15}N$ proxy in the terrestrial environment, we hypothesize two conditions. First, they must persist in quantities sufficient for isotopic analysis. Second, they must retain the $\delta^{15}N$ value of the leaves they derived from, with at most a constrainable isotopic offset, throughout the processes of biosynthesis, senescence, and decomposition--regardless of changes in environmental conditions.

We evaluate these hypotheses on the natural climate gradient of Kohala Mountain, on the Big Island of Hawaii. These ecosystems are well suited for testing questions about relationships between climate, biogeochemistry, and the preservation or degradation of organic compounds. Relatively low plant diversity and broad environmental niches permit comparison of the bulk and compound-specific $\delta^{15}N$ of leaves, litter, and soil, of similar grassland communities in sites ranging widely in climate

and $\delta^{15}N$, to investigate patterns of deviation of compound-specific values from bulk values.

## 2. Methods

### 2.1 Site Description

The four sites of this study are located on Kohala Mountain, the oldest of five volcanoes that make up the Big Island of Hawaii (Fig. 1). The sites are a subset of a well characterized climosequence running from the crest of Kohala down the leeward side

(see (Chadwick et al., 2003), (Kelly et al., 1998b), (Vitousek and Chadwick, 2013), (von Sperber et al., 2017)). Prevailing winds from the northwest create a dramatic gradient in precipitation, from 2500 mm falling annually at the uppermost site to 210 mm at the lowest site (Giambelluca et al., 1986). In contrast to the substantial shift in precipitation, the elevation of the mountain produces only a moderate gradient in air temperature, ranging from 17°C at the upper site to 23°C at the lowest site.





Soils have been forming since emplacement of the Hawi lava flows, approximately 150,000 years ago (Chadwick et al., 2003;Spengler and Garcia, 1988;Wolfe and Morris, 1996). This gradient thereby brings into focus precipitation as the most substantial systematic facto of change across sites.

Climate history for this region has been reconstructed from pollen assemblages in cores from bogs near the Kohala summit (Hotchkiss, 1998), and inferred from patterns of sea-level change (Ziegler et al., 2003) Aeolian deposits (Porter, 1997), and soil calcite deposits at the drier sites (Chadwick et al., 2003;Porter, 1997). During glacial periods the summit was cooler and drier, while the lower sites show less climate variability, having remained arid throughout soil development.

Vegetation across the gradient was altered by clearing of land for pasture in the last two hundred years (Cuddihy and Stone, 1990;Chadwick et al., 2007), resulting in the introduction of non-native species and grasses with the C4 carboxylation pathway. Site C is lowland dry scrubland and grassland (150 to 500 mm annual rainfall), dominated by buffel grass (*Cenchrus ciliaria*) and the leguminous tree keawe (*Prosopis pallida*). Sites F (790 mm annual rainfall) and I (1260 mm annual rainfall) lie in lowland dry and mesic forest, woodland, and shrubland zones, which through conversion to pasture are dominated by the grass
kikuyu (*Pennisetum clandestinum*). Site M lies in the wet forest and woodland zone (2500 mm annual rainfall), in which ohia (*Metrosideros polymorpha*) is the dominant tree and the native understory was previously dominated by tree fern (*Cibotium spp.*) and uluhe (*Dicranopteris linearis*), but conversion of land to pasture has introduced kikuyu and other species. The herbs clover (*Desmodium incanum*) and Madagascar ragwort (*Senecio madagascariensis*) were additionally prevalent at sites I and M.

All pits were dug to at least 50 cm depth in open, grassy areas, with minimal slope. All of the sites in this study experience grazing by cows, with sites F and I grazed most heavily.

## 2.2 Sample collection

Leaves were sampled across sites from six plant species; from the upper 1/3 of the canopy in the case of the trees: Two grasses, *P. clandestinum* and *C. ciliaria*; two herbs, *D. incanum* and *S. madagascariensis*; and two trees, *P. pallida* and *M. polymorpha*. Two of these species are N fixers (*D. incanum and P. pallida*), and two have a C4 carboxylation pathway (*P. clandestinum and C. ciliaria*) while the others have C3. Each species was collected wherever present within a radius of ~50 m from the soil pit dug at each site, but not all species were present at all sites. For grasses and herbs, at least three individuals (and generally
many more) were collected at each site and bulked for processing as a single sample. For trees, leaves were collected from at least 3 branches per tree and bulked for processing.



Soils were sampled from pits dug to a depth of greater than 50 cm. At these sites, litter and organic horizon layers were sampled as a single genetic horizon termed "litter," consisting of organic materials with fibrous through humic composition and which had a total thickness of less than 2 cm. Mineral horizons were sampled at regular depth intervals of ~10 cm to a depth of 50 cm. Samples were bagged in Whirlpak bags and kept dark and chilled on ice in coolers until processing.

## 2.3 Sample preparation

Live foliage was rinsed with deionized (DI) water to remove dust or other contaminants and then freeze-dried for preservation. Soils and litter were likewise freeze-dried. Dried samples were ground using either a carbide-steel shatter box (SPEX SamplePrep, University of California, Davis) or a mortar and pestle. Samples were stored in a dark freezer.

## 2.4 Compound extraction and purification

Samples were handled under very limited light conditions to minimize photo-degradation. Pigments were extracted by sonication in triplicate using ~100 mL of acetone (300 ml total) for 1 to 80 g of subsamples. Extracts were concentrated by evaporation to near dryness under nitrogen or argon. The condensed acetone extract was partitioned between 10 ml hexane
and 30 ml Milli-Q water via liquid-liquid extraction to remove more polar compounds. The hexane fraction was retrieved, and dried under gentle flow of argon gas, and this crude chlorophyll extract was dissolved in dimethylformamide, and passed through a syringe filter (0.4μm) to remove particulates prior to High Pressure Liquid Chromatography (HPLC) chromatography.

Two-dimensional HPLC was used to separate and purify chlorophyll *a* and *b* and their photo-reactive degradation products. Each individual compound was collected following separation with a fraction collector system of the HPLC (Agilent 1200 Series with a diode-array detector (DAD), and a fraction collector). For the first HPLC separation step using reversed phase, the sample was passed through a Zorbax Eclipse XDB C18 column (9.4 x 250 mm; 5 μm) with a liquid phase consisting of acetonitrile, ethyl acetate, and pyridine in variable ratio, for 35 minutes at 4.2 ml/min and 30°C, after (Kusch et al., 2010). For
all samples, pheophytin *a* (pheo *a*) was collected upon elution at 19 minutes. For five plant samples, chlorophyll *a* (chl *a*) was collected upon elution at 12 minutes. Spectra were checked for purity across wavelengths 200-900 nm. In the second HPLC separation step, also using reverse phase, the sample was passed through an Agilent Zorbax Eclipse PAH column (4.6 x 250 mm; 5 μm), with a liquid phase consisting of acetonitrile, ethyl acetate, and pyridine in variable ratio, for 35 min at a flow rate of 1 ml/min and 15°C.



## 2.5 Analytical methods

Nitrogen isotopic composition, total N, and total C analysis of bulk soil (including litter) and leaf samples was performed using an elemental analyser interfaced to a continuous-flow isotope ratio mass spectrometer (EA-IRMS) at the University of California, Davis Stable Isotope Facility. The mean value of analytical precisions obtained for standard materials is 0.3‰ for

$\delta^{15}N$. C/N ratios are reported in mass units. In the case of soil, analysis was performed on ~20 mg of material using an Elementar Vario EL Cube or Micro Cube elemental analyser (Elementar Analysensysteme GmbH, Hanau, Germany) interfaced to a PDZ Europa 20-20 isotope ratio mass spectrometer (Sercon Ltd., Cheshire, UK). In the case of leaves, analysis was performed on ~4 mg of material using a PDZ Europa ANCA-GSL elemental analyser interfaced to a PDZ Europa 20-20 isotope ratio mass spectrometer (Sercon Ltd., Cheshire, UK). Samples are combusted at 1000°C in a reactor packed with

copper oxide and lead chromate. Following combustion, oxides were removed in a reduction reactor (reduced copper at 650°C). The helium carrier then flowed through a water trap (magnesium perchlorate). $N_2$ and $CO_2$ were separated using a molecular sieve adsorption trap (for soil) or Carbosieve GC column (65°C, 65 mL/min) (for leaves) before entering the IRMS. Bulk isotopes of two plants (Koh F buffel/kikuyu) and one soil core (4 layers, fresh and litter1) were analysed at JAMSTEC.

$\delta^{15}N$, total N, and total C analysis of isolated pheo $a$ fraction was performed on a Nano-EA-IRMS at the Japan Agency for Marine Earth-Science and Technology (JAMSTEC). This is a modification of an EA-IRMS system to achieve ultra-sensitive analysis (Ogawa et al., 2010). All samples contained at minimum 10 nmol N, resulting in $\delta^{15}N$ precision of $< \pm0.4‰$. To investigate the possibility of N contamination, nitrogenous volatile was analysed for the pheophytin $a$ fraction by gas chromatography/nitrogen-phosphorus detector (GC/NPD) with trimethylsilyl derivatization (BSTFA, Agilent Technologies,

Palo Alto, CA) at JAMSTEC. C/N ratios from EA/IRMS were used as the "purity indicator" of each chlorophyll compound. As is described below, if a sample showed the clean spectrum pattern of a chlorin but had significantly large C/N ratio, it is likely that it contained C-containing contaminants, such as carbon hydrates, which are not detectable by DAD.

## 2.6 Purity of isolated compounds

While absorption spectra in UV/visible wavelengths showed no evidence for other absorptive components in the pheo $a$ extracts, C/N weight ratios greater than 11.8 revealed inclusion of non-pheo $a$ carbon in the extracts. C/N ratios were higher in pheo $a$ extracted from litter (31) than plant samples (14.5), and even higher in pheo $a$ samples from soils (113). Lack of relationship between C/N and bulk-pheo $\delta^{15}N$ offset suggests that impurities do not contribute significantly to measured $\delta^{15}N$ values, however (Fig. 2). This supposition was confirmed by results from the GC-NPD), which showed a lack of nitrogenous

compounds (e.g. containing amino groups) in the pheo $a$ extracts. Although we caution that polar and less-volatile compounds cannot be detected by this method, the extraction methods make amino acids and other nitrogenous contaminants unlikely.



The analytical protocol is designed to remove as many of such contaminating compounds as possible; e.g. amino acids are not soluble in acetone.

### 2.7 Isotopic notation

Nitrogen isotopic compositions are reported using conventional delta (δ) notation:

$$\delta\ (‰) = (R_{sample}/R_{standard} - 1) \times 1000 \qquad \text{(Equation 1)}$$

where $R$ represents the $^{15}N/^{14}N$ ratio and subscripts indicate the sample or isotopic reference. The sample isotopic composition
is measured directly relative to the $N_2$ laboratory reference gas ($\delta S_{Ref}$), and the composition of the sample relative to the internationally recognized $\delta^{15}N$ reference, AIR, is calculated by:

$$\delta S_{AIR} = \delta S_{Ref} + \delta Ref_{AIR} + 10^{-3}\delta S_{Ref}\ \delta Ref_{AIR} \qquad \text{(Equation 2).}$$

Nitrogen isotopic fractionation of pheo $a$ relative to bulk leaf tissue ($^{15}\varepsilon_{pheo\text{-}bulk}$) and chlorophyll a relative to bulk leaf tissue ($^{15}\varepsilon_{chl\text{-}bulk}$) are defined according to:

$$^{15}\varepsilon_{compound\text{-}bulk} = 1000\left[\frac{\delta15Ncompound + 1000}{\delta15Nbulk + 1000} - 1\right] \qquad \text{(Equation 3).}$$

**3.  Results**

### 3.1 Bulk isotope and C and N concentration data

Site-averaged bulk $\delta^{15}N$ of all soil samples decreased with increasing precipitation (and elevation) across all sites, from an average of 12.4‰ at site C to 5.1‰ at site M (Table 1). Site-averaged $\delta^{15}N$ of litter decreased between sites C and M (4.3‰ to 2.8‰), but sites F and I were higher than either of these values (9.0‰ and 7.2‰). Average foliar $\delta^{15}N$ likewise decreased
between sites C and M (4.2‰ to 0.5‰), but site F (9.1‰) was higher than C (Fig. 3). Site-averaged bulk soil $\delta^{15}N$ values for those samples on which $\delta^{15}N_{pheo}$ was measured follow similar trends (Table 2), though on average site values are slightly lower (8.4 vs. 8.9‰), reflecting the shallower average depth of the soil samples.





Soil %N increased from site C (0.2%) to site M (1.2%), and %C increased across these sites from 1.9% to 17.2%. C/N increased between these sites from 11.5 to 16.3 (Table 1). Litter %C and C/N was notably higher at site M (36.3% and 25.2) than at the other sites, although litter %N was relatively flat across all sites. Vegetation %C was likewise highest at site M (43.1%), though this value was less exceptional compared with vegetation at other sites.

Vegetation had the highest C/N (average of 19.9), followed by litter (17.2) and soil (12.4). At sites C, F, and M, C/N of litter is closer to that of vegetation than to soil, while at site I C/N of litter is closer to that of soil than to vegetation.

### 3.2  Chlorin compound detection

Chlorins were detected in leaf, litter, and soil samples, including in soil mineral horizons up to a depth of 32 cm (Table 2). As

chlorophyll *a* has greater absorbance at 660 nm than Pheo *a,* direct proportionality between relative HPLC peak areas and relative compound abundance in the sample should not be assumed. In plant leaves, chlorophyll *a* was the dominant chlorin. Chlorophyll *a* was also found in smaller to trace amounts in litter and some soils. In litter, pheo *a* was the dominant pigment, with the exception of site M, where chlorophyll *a* was more abundant in the litter sample. In site C, chlorophyll *a* was absent from litter and soil. In soils, pheophytin *a* was the most abundant degradation product. Pheo *a*, targeted for isotopic analysis

due to its superlative abundance, was present in sufficient concentration for isotopic analysis above ~20 cm in soils at most sites.

### 3.3  Pheophytin *a* N isotope data  of leaves, litter, and soil

*Intra-leaf isotope offsets*—Across all plant samples, $\delta^{15}N_{pheo}$ of live foliage was significantly, linearly correlated with bulk $\delta^{15}N_{leaf}$ (slope = 0.9; y-intercept = −1.1; adjusted $R^2$ = 0.8; p = 0.000002507) (Fig. 4). $^{15}\varepsilon_{pheo\text{-}bulk}$ was equal to 1.4‰ (± 2.3‰)

across all plant samples (Eq 1).

Of the six sampled species, all but the two trees exhibited mean $^{15}\varepsilon_{pheo\text{-}bulk}$ values ≤ 1.5‰ (Table 3): $^{15}\varepsilon_{pheo\text{-}bulk}$ for *P. pallida* was 6.5‰ and $^{15}\varepsilon_{pheo\text{-}bulk}$ for *M. polymorpha* was 2.5‰. If the *P. pallida* sample is omitted, $^{15}\varepsilon_{pheo\text{-}bulk}$ drops to 0.71‰ (±1.3‰).

$^{15}\varepsilon_{pheo\text{-}bulk}$ was largest at site C (4.0‰) and smallest at site F (0.12‰). For a given species, average $^{15}\varepsilon_{pheo\text{-}bulk}$ tended either to remain flat or slightly decrease into the wettest sites (Fig. 5).

*$\delta^{15}N_{pheo}$ offsets across leaf-litter-soil*—$\delta^{15}N$ of Pheo *a* in litter was on average 0.3‰ higher than the $\delta^{15}N$ of pheo *a* of live foliage at a common site. Average difference between the $\delta^{15}N$ of bulk litter and foliage were slightly higher, at ~ 2.6‰ at a

common site (Table 4). Pheo *a*-specific soil $\delta^{15}N$ values were on average 1.3‰ higher than pheo *a* litter values at a common site; bulk $\delta^{15}N$ soil values are 2.6‰ higher than bulk litter. The average offset between $\delta^{15}N_{phe}$ of soil and live foliage at a site was 1.1‰; the average offset between bulk soil and bulk vegetation $\delta^{15}N$ was 4.9‰.



### 3.4 Soil depth profiles

Soil $\delta^{15}N_{bulk}$ was, on average, higher than the $\delta^{15}N_{bulk}$ of overlying litter, and there was a slight trend of increasing $\delta^{15}N_{bulk}$ with increasing depth in the upper ~25 cm of soil pits (Fig. 6). $\delta^{15}N_{pheo}$ of soil also displayed higher values relative to overlying

5  litter in sites F and I and in part of the profile at site M (Fig. 6). Soil $\delta^{15}N_{bulk}$ values returned to slightly more negative values deep in the profiles; particularly notable at site F. At site C, $\delta^{15}N_{pheo}$ along the soil profile not deviate significantly from that of the overlying litter. At site M, $\delta^{15}N_{pheo}$ values increased slightly with depth in the upper profile, but then decreased while $\delta^{15}N_{bulk}$ steadily increased with greater depth. In sum, $\delta^{15}N_{pheo}$ of soil did not follow $\delta^{15}N_{bulk}$, nor did it constantly track $\delta^{15}N_{pheo}$ of modern plants at a common site.

### 4.  Discussion

Nitrogen isotopic offsets between pheophytin $a$ and live foliage ($^{15}\varepsilon_{pheo-bulk}$) are generally small (mean of 1.4‰) and well constrained (s.d. ± 2.3‰, Fig. 3) across our sites, marking $\delta^{15}N_{pheo}$ as a useful proxy for bulk $\delta^{15}N_{leaf}$. The robustness of $^{15}\varepsilon_{pheo-bulk}$ across the wide range in environmental variables and values of $\delta^{15}N_{leaf}$ observed across these sites (Fig. 4) suggests that

isotope effects from species or physiological changes will be relatively small sources of variation in $\delta^{15}N_{pheo}$ compared with changes to primary $\delta^{15}N_{leaf}$. For example, average community foliar $\delta^{15}N_{leaf}$ varies by about ~10‰ across terrestrial biomes in the modern environment (Craine et al., 2009), and paleo records have inferred comparable shifts in $\delta^{15}N_{leaf}$ of 10‰ between glacial-interglacial cycles (Stevens et al., 2008). The possible effect of species shifts, and particularly growth of forests, should be considered when evaluating proxy $\delta^{15}N_{pheo}$ records, however. One species, *P. pallida*, has $^{15}\varepsilon_{pheo-bulk}$ of 6.5‰. The other tree

studied here, M. *Polymorpha*, had the next largest $^{15}\varepsilon_{pheo-bulk}$ (2.5‰), while herbaceous plant values were all considerably smaller (-0.29 to 1.4‰), raising the question of whether there may be a positive isotope effect of cellulose accumulation on $\delta^{15}N_{pheo}$.

Pheo $a$ was present in quantities sufficient for isotopic analysis in litter and the uppermost mineral soil sample at all sites, and

down to 30 cm at the wettest site (Fig. 4), inviting exploration of soil $\delta^{15}N_{pheo}$ as a proxy for $\delta^{15}N_{leaf}$. $\delta^{15}N_{pheo}$ of soil and litter match $\delta^{15}N_{pheo}$ of overlying leaves at only one studied site (site C). Two alternatives could explain the lack of coherence in leaf, litter, and soil pools across most sites: 1) there is isotopic alteration during senescence and mineralization of chlorophyll, and 2) litter and soil pheo $a$ pools have other sources than the plants we sampled, either on the current landscape, or from previous vegetation covers.

To investigate the expression of fractionation on demetallation of chlorophyll in these samples, we can compare the $\delta^{15}N_{pheo}$ with chlorophyll $a$ (chl $a$) from the same sample. Chl $a$ was only in sufficient abundance for isotopic measurement in live plant

leaves. Chl $a$ is about 0.05‰ depleted in $^{15}$N relative to bulk leaves in this study. This is consistent with previous studies, which found chlorophyll in plants to be 1.2‰ depleted in $^{15}$N relative to bulk leaf tissue (Kennicutt et al., 1992;Chikaraishi et al., 2005). Pheo $a$ is accordingly about 2‰ enriched in $^{15}$N relative to chl $a$ in our leaf samples, which could point to fractionation either within the leaf or in the laboratory (Sachs, 1997). Because chlorophyll abundance is greatly reduced

between leaves and litter, any fractionation is likely to be expressed as a difference between leaves and litter, not as a difference between litter and soil or within soil. Fractionation on demetallation from chlorophyll should therefore not account for changes in $\delta^{15}N_{pheo}$ in the soil.

Soil is enriched in $^{15}$N relative to litter at our sites, consistent with observations from a wide range of environments, including

temperate rainforest (Menge et al., 2011), temperate deciduous forests (Templer and Dawson, 2004), boreal forests (Hyodo and Wardle, 2009), tropical forests (Martinelli et al., 1999), temperate grasslands (Baisden et al., 2002;Brenner et al., 2001), and elsewhere (Hobbie and Ouimette, 2009). Because post-depositional processing of N involves the preferential loss of $^{14}$N from the soil N pool without replacement, through removal of the products of mineralization and denitrification, bulk soil $\delta^{15}$N is highly dynamic and will tend to increase along a decomposition gradient. In contrast, if there is no fractionation on

breakdown of pheo $a$, $\delta^{15}N_{pheo}$ in soil should not depart from the $\delta^{15}N_{pheo}$ of litter inputs.

In soil profiles across the climosequence, $\delta^{15}N_{bulk}$ and $\delta^{15}N_{pheo}$ do not exhibit similar depth patterns (Fig. 6). This suggests that different processes govern these two records in soil. While $\delta^{15}N_{bulk}$ values drift increasingly positive across the leaf-litter-soil continuum, pheo $a$ values remain centered around site average leaf values, and sometimes show slight decreases in value from

leaf to litter and soil (Table 4). This suggests that mixing of inputs may account for litter and soil pheo $a$ $\delta^{15}$N values, while fractionating losses from the bulk soil N pool are required to explain $\delta^{15}$N bulk values.

$\delta^{15}N_{pheo}$ of litter is a window into possible isotopic effects of decomposition on $\delta^{15}N_{pheo}$ of plant leaves. The difference between $\delta^{15}N_{pheo}$ of leaves and litter averaged close to zero and was substantially smaller than the difference in $\delta^{15}N_{bulk}$ (0.3‰ *versus*

2.3‰), supporting greater isotopic fidelity along the decay continuum in pheo $a$ $\delta^{15}$N than bulk $\delta^{15}$N. Some of the site variability in isotopic offsets between litter and leaves/soil is likely due to differences in how well decomposed the collected litter was from site to site: at sites C, F, and M, C/N of litter is closer to that of vegetation than to soil, while at site I C/N of litter is closer to that of soil than to vegetation.

It is likely that litter samples do not reflect equal contributions from the plants sampled at the same site. Litter was collected from the surface of the soil pits, while vegetation was sampled from a radius of many meters, and plant taxa were not identified in litter samples to allow for source attribution. Soil algae and bacteria are considered insignificant contributors to soil pheo $a$, as the absorbance spectra of soil extracts showed that chlorophyll degradation products are dominated by higher plant contributions.



Pheophytin has two pathways of generation: Like chlorophyll *a*, it is biosynthesized in plant leaves from glutamate where it serves as an electron acceptor in photosystem II. Additionally, it is a product of chlorophyll degradation, whether via senescence, grazing, or decomposition across the leaf-soil continuum, in which the central Mg is replaced by two H atoms.

Transformation from chlorophyll to pheophytin involves breaking N-Mg bonds, and so has potential for fractionation of N isotopes. However, the N-Mg bond is not a normal covalent bond, but a bond loosely connecting a ligand and metal in complex. Because the energy needed to break this bond is substantially smaller than the covalent bond, Mg loss from Chl would theoretically be expected to have little or no N isotope fractionation. Senescence is unlikely to impart significant fractionation from bulk leaves given the observation that bulk leaf $\delta^{15}N$ does not change on abscission (Kolb and Evans, 2002) and the

understanding that N contained in the tetrapyrrole structure of chlorophyll is not recycled by the plant (Eckhardt et al., 2004). However, fractionation on demetallation of chlorophyll to pheophytin has been observed in a laboratory setting with an effect of up to 2‰ (Sachs, 1997). Mineralization of pheo *a* is unlikely to alter $\delta^{15}N$ of the remaining pheo *a* pool because known products of chlorin defunctionalization retain the four atoms of tetrapyrrole N and their associated bonds (Keely, 2006).

It is unlikely that environmental changes that took place between production of litter and growth of current leaves explain the observed differences between $\delta^{15}N$ of plants and litter, given that litter at these sites is estimated to be no more than a few years old. Soil pools, however, could contain pheo *a* that is considerably older. There is reason to expect that deeper pheo *a* will be older than overlying pheo *a* in soil profiles, and increasing recognition that compounds can be preserved in the soil matrix much longer than their inherent lability would predict (Mikutta et al., 2006;Torn et al., 1997;Marin-Spiotta et al.,

2011;Kramer et al., 2012).

While we do not know the age of the sampled soil pheo *a,* radiocarbon dates of 4130 and 8030 yBP taken on soil organic carbon (SOC) deep within the soil profiles at other sites along the climosequence suggest that the pheo *a* compounds may exhibit a range of several thousand years over the depths of these profiles (Chadwick et al., 2007). Bulk soil carbon isotope

values suggest that C4 pasture grasses, introduced over the last two hundred years, have been incorporated in the SOC in the top 40 cm of soil on these sites (Kelly et al., 1998a).

Patterns of radiocarbon ($\Delta^{14}C$) depletion of SOC in profile (Baisden et al., 2002;Townsend et al., 1995;Tonneijck et al., 2006;Trumbore, 2009) indicate that SOC age increases with soil depth. Departures from this trend have been explained by the

introduction of modern material by plant roots (Baisden et al., 2002), but as chl *a* and pheo *a* are photosynthetic pigments, we expect that inputs to the soil originate exclusively as litter accumulated at the soil surface, and for age of the chlorophyll derivatives to therefore increase with depth. Bioturbation and episodic leaching could be disruptive of space-for-time trends, but based on the pattern observed in bulk SOC $\Delta^{14}C$, are unlikely to interfere with millennial scale patterns given appropriate vertical sampling distances. In fact, in aggrading profiles, bioturbation has been shown to contribute to the increase in SOC



age with depth, by transporting SOC downward over short distances and migrating upwards as soil accumulates overhead (Tonneijck and Jongmans, 2008).

If down-profile $\delta^{15}N_{pheo}$ values did represent prior landscape $\delta^{15}N_{leaf}$ values, these data suggest greater changes to the N cycle
at sites F and I than C and M, which could reflect heavier grazing at sites F and I. Climate has been more constant throughout the Holocene at lower elevations than higher elevations on Kohala (Hotchkiss, 1998), which could further account for the relatively constant $\delta^{15}N_{pheo}$ values at site C.

If the total soil $\delta^{15}N$ pool reflects fractionation on loss of N, but the soil $\delta^{15}N_{pheo}$ pool does not, we expect that the more
important gaseous losses are relative to leaching losses at a site, the higher $\delta^{15}N_{bulk}$ and the greater the offset between $\delta^{15}N_{pheo}$ and $\delta^{15}N_{bulk}$ will be. Environments where denitrification is more important relative to leaching tend to be dry environments, due to much smaller isotope effects of non-gaseous losses (Houlton and Bai, 2009), and drier sites do indeed have higher $\delta^{15}N$ in this study. Differences between $\delta^{15}N_{pheo}$ and $\delta^{15}N_{bulk}$ do not show clear trends along the climosequence, however. This could be due to the complexity of soil moisture at these sites, reflected in the variability in down-profile soil bulk $\delta^{15}N$.

Direct testing of the hypothesis of increasing age with depth of pheo *a* would be enabled by improving the purification method of pheophytin *a* from soil. Removing contaminating C would make stable or radiocarbon of pheo *a* available tools to shed light on sources of pheo *a* in soil profiles, age of the compounds, and with what resolution change in N cycling over the temporal window provided by the chlorins depth profile is observable (Ishikawa et al., 2015). These questions could also be
averted by measuring $\delta^{15}N_{pheo}$ in soil profiles with constrained ages, such as buried horizons.

**Conclusions**

$\delta^{15}N_{pheo}$ in leaves provides a molecular recorder of foliar $\delta^{15}N$, and provides a means to trace leaf nitrogen signatures in litter and soils. The compound is found in litter and upper soil horizons in abundance sufficient for isotopic analysis, where it shows greater fidelity to leaf $\delta^{15}N$ than does bulk material. $\delta^{15}N_{pheo}$ of soil may reveal temporal changes in nitrogen cycle behavior—
e.g. the availability of nitrogen to plants, and whether nitrogen losses from the ecosystem had a dominant atmospheric or hydrologic fate. Due to the well-constrained photoautotrophic sources of chlorins, their lack of confounding heterotrophic enrichment, and their minimum N isotope fractionation during burial processes, CSIA of chlorins offers several advantages over bulk isotope analysis on the study of soil C and N cycles. $\delta^{15}N_{pheo}$ of soil is a most promising proxy for $\delta^{15}N_{leaf}$ where organic matter inputs are high and profiles aggrade. The potential to investigate decadal-to-millennial scale N cycle dynamics
will depend largely on conditions for preservation of soil organic matter.



**Data availability**

The data can be accessed by email request to the corresponding authors.

**Author Contributions**

SG and BH acquired funding for the project. SG, BH, and OC collected samples. NO, KF, and BH contributed laboratory
space and materials. SG, NO, KF, YC, NO, and HS collaborated on laboratory methodology; SG, NO, HS, and YC conducted
analyses of samples. SG performed data analysis, generated figures, and wrote the initial draft; BH, NO, KF, YC, HS, and OC
contributed to the manuscript.

**Competing interests**

The authors declare that they have no conflict of interest.

**Acknowledgements**

Funding for this study was provided to SG by a United States Environmental Protection Agency Science To Achieve Results
(STAR) Graduate Fellowship, a National Science Foundation (NSF) Doctoral Dissertation Improvement Grant, a joint NSF
East Asia and Pacific Islands Summer Institute and Japan Society for the Promotion of Science Fellowship, an NSF Inter-
University Training for Continental Ecology Award, a University of California at Davis (UCD) Henry A. Jastro Graduate
Student Award, a UCD Graduate Studies Travel Grant, and through an NSF Career Grant to BH.

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

**Figures**

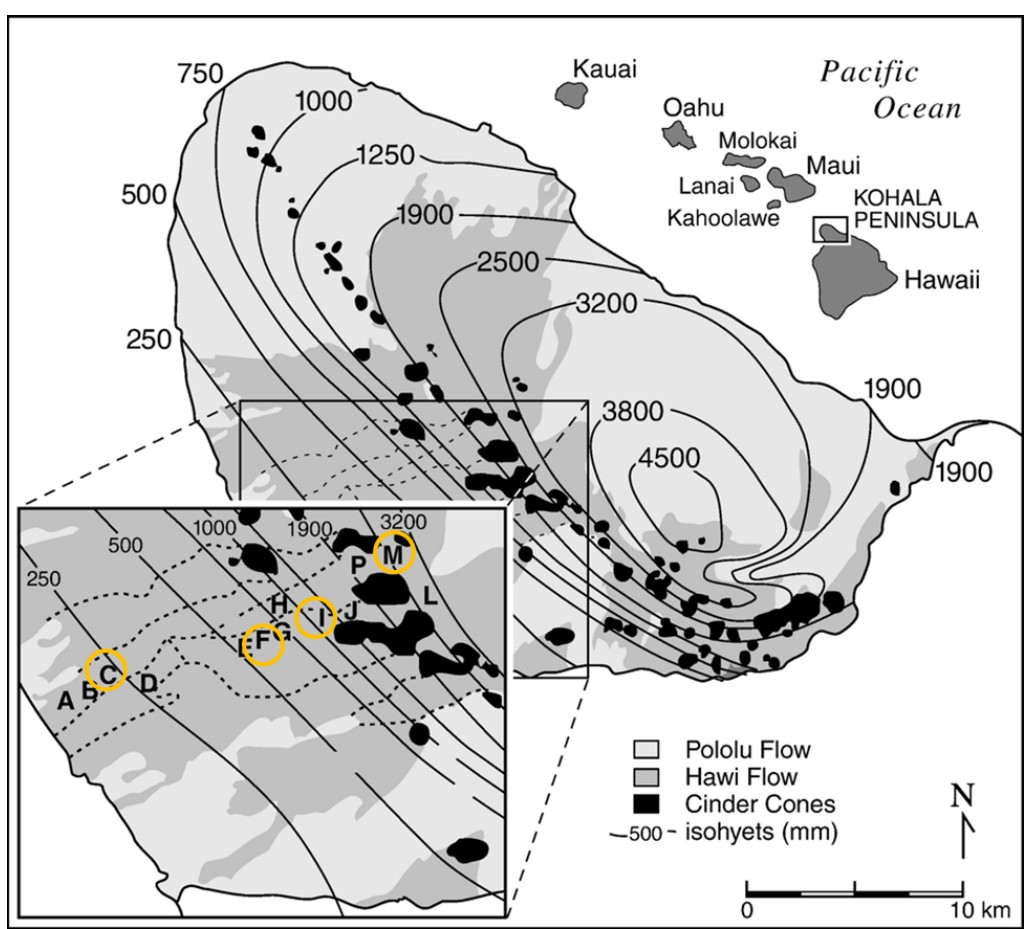

**Figure 1:** Map of Kohala climosequence sites with respect to rainfall isoclines, reproduced from Chadwick et al. (2007). Sites sampled for this study (C, F, I, and M) are circled in orange.







**Figure 2:** C/N ratios for pheo *a* isolates higher than 11 (dashed line) reveal contamination by carbon compounds. Effect of this contamination on pheo a $\delta^{15}$N is investigated by comparing "pheo *a*" molar C/N with the offset between pheo *a* and bulk $\delta^{15}$N for litter (black), soil (red), and vegetation (green) samples. Lack of a systematic relationship between high C/N ratios and greater similarity of pheo *a* $\delta^{15}$N to bulk values is evidence against contamination of the pheo *a* samples by N from the greater bulk sample.



**Figure 3:** Patterns in isotopes of bulk material (open circles) and pheo *a* extracts (closed triangles) from plant leaves (Veg, green), litter (black), and soil (red). Deep soils (blue) have a depth of greater than 30 cm and only a bulk isotope measurement was made for these samples. Lines connect site averages between bulk (solid) and pheo *a* (dashed) measurements.



**Figure 4:** $\delta^{15}$N of pheo *a* (green) and chlorophyll a (orange) correlates linearly with bulk leaf $\delta^{15}$N. The linear model for $\delta^{15}$N of pheo *a* as a function of bulk leaf $\delta^{15}$N is printed as a dashed line. A line with a slope of 1 (solid line) is plotted for reference. Each sample's species is identified with a label: CC = C. ciliaria*, DI = D. incanum†, MP= M. polymorpha‡, PC= P. clandestinum*, PP = P. pallida†‡, SM = S. madagascariensis.*=C4 pathway, †=N fixer, ‡=tree







**Figure 5:** Comparison of species average intra-leaf $\delta^{15}N$ offset between pheo *a* and bulk ($^{15}\varepsilon_{\text{pheo-bulk}}$) across sites. CC = C. ciliaria*, DI = D. incanum†, MP= M. polymorpha‡, PC= P. clandestinum*, PP = P. pallida†‡, SM = S. madagascariensis.*=C4 pathway, †=N fixer, ‡=tree



**Figure 6:** Bulk samples (open circles) and pheo *a*-specific (closed triangles) $\delta^{15}$N in depth profiles across the rainfall gradient from dry (C) to wet (M) sites. Dashed lines set apart the vegetation samples (top) from litter samples (middle) and soil samples (bottom). The surface of the mineral soil is at a depth of 0.





**Tables**

|  | C | F | I | M | Site average |
|---|---|---|---|---|---|
| $\delta^{15}N_{bulk}$ leaf | 4.15 | 9.14 | 0.39 | 0.46 | 3.54 |
| $\delta^{15}N_{bulk}$ litter | 4.29 | 8.95 | 7.18 | 2.8 | 5.8 |
| $\delta^{15}N_{bulk}$ soil | 12.39 | 9.91 | 8.21 | 5.13 | 8.91 |
| soil %C | 1.93 | 9.67 | 10.13 | 17.24 | 9.74 |
| soil %N | 0.17 | 0.83 | 0.95 | 1.19 | 0.78 |
| soil C/N | 11.51 | 11.31 | 10.62 | 16.25 | 12.42 |
| litter %C | 21.54 | 29.52 | 18.67 | 36.28 | 26.5 |
| litter %N | 1.38 | 1.94 | 1.48 | 1.44 | 1.56 |
| litter C/N | 15.66 | 15.24 | 12.61 | 25.17 | 17.17 |
| veg %C | 41.48 | 37.06 | 39.92 | 43.11 | 40.39 |
| veg %N | 2.69 | 2.65 | 2.32 | 2.28 | 2.49 |
| veg C/N | 15.5 | 14.99 | 20.92 | 28.18 | 19.9 |

**Table 1**: Site averages, for all bulk samples. Percent C and N and C/N are weight-based.



| Site | Name | Chl $b$ | Chl $a$* | Pheo $b$ | Pheo $a$ | Sum All | %Pheo $a$ | %Chl $a$* |
|------|------|---------|----------|----------|----------|---------|-----------|-----------|
| C | C. *ciliaria* | 22119 | 81074 | 39 | 52545 | 260225 | 44 | 62 |
| | Litter/O | | | | 5173 | 9622 | 54 | 0 |
| | 00-16 | | | | 3225 | 5178 | 62 | 0 |
| F | P. *clandestinum* | 3980 | 36469 | | 21452 | 75487 | 28 | 48 |
| | Litter/O | 525 | 5496 | | 4625 | 14995 | 31 | 37 |
| | 03-08 | | 559 | | 2483 | 4391 | 57 | 13 |
| | 08-13 | | 1104 | | 1590 | 3687 | 43 | 30 |
| I | S. *madagascariensis** | 15264 | 92967 | | 4106 | 144176 | 3 | 64 |
| | Litter/O | | 855 | | 2820 | 10036 | 28 | 9 |
| | 00-10 | | | 225 | 1058 | 1711 | 62 | 0 |
| M | D. *incanum* | 1146 | 9385 | | 2432 | 16299 | 15 | 58 |
| | Litter/O | 276 | 1260 | | 963 | 2952 | 33 | 43 |
| | 00-15 | | | 2357 | 7437 | 13698 | 54 | 0 |
| | 24-32 | | | 88 | 332 | 656 | 51 | 0 |

**Table 2:** Relative compound abundance in a plant sample and the litter and soil horizons on which $\delta^{15}N_{pheo}$ was measured, grouped by site. Values reported are summed peak areas of absorbance at 660 nm in units of mAU, summed for all collected injections of the sample. Values are first reported for individual compounds and then compared with the summed area of all detected chlorin peaks, first as absolute values and then as a percentage of chlorin peak area. *Chl $a$ and Py Chl $a$ are combined.

| | mean ε | min ε | max ε | range | n |
|---|--------|-------|-------|-------|---|
| C. ciliaria* | 0.81 | -0.45 | 1.54 | -1.99 | 3 |
| P. clandestinum* | -0.29 | -2.06 | 1 | -3.06 | 5 |
| P. pallida†‡ | 6.53 | 6.39 | 6.68 | -0.29 | 2 |
| D. incanum† | 0.54 | 0.34 | 0.73 | -0.39 | 2 |
| S. madagascariensis | 1.44 | 0.58 | 2.31 | -1.73 | 2 |
| M. polymorpha‡ | 2.48 | 2.26 | 2.71 | -0.46 | 2 |

**Table 3:** Average, minimum, maximum, and range of intra-leaf $\delta^{15}N$ offsets between pheo $a$ and bulk ($^{15}\varepsilon_{pheo\text{-}bulk}$) for n measurements of each sampled species across all sites. *=C4 pathway, †=N fixer, ‡=tree



|  | C | F | I | M | Site average |
|---|---|---|---|---|---|
| $\delta^{15}N_{bulk}$ leaf | 4.15 | 9.14 | 0.68 | 0.04 | 3.5 |
| $\delta^{15}N_{phe}$ leaf | 8.15 | 9.43 | 1.2 | 0.77 | 4.89 |
| $\delta^{15}N_{bulk}$ litter | 4.29 | 8.95 | 7.18 | 2.8 | 5.8 |
| $\delta^{15}N_{phe}$ litter | 8 | 4.7 | 5.1 | 0.7 | 4.62 |
| $\delta^{15}N_{bulk}$ soil | 10.37 | 10.77 | 7.19 | 5.35 | 8.42 |
| $\delta^{15}N_{phe}$ soil | 8.1 | 7.75 | 6.4 | 1.65 | 5.97 |
| $\varepsilon$ phe-leaf | 3.99 | 0.28 | 0.52 | 0.73 | 1.44 |
| $\Delta_{phe}$ veg-litter | 0.15 | 4.73 | -3.9 | 0.07 | 0.26 |
| $\Delta_{bulk}$ veg-litter | -0.14 | 0.2 | -6.5 | -2.76 | -2.3 |
| $\Delta_{phe}$ litter-soil | -0.1 | -3.05 | -1.3 | -0.95 | -1.35 |
| $\Delta_{bulk}$ litter-soil | -6.08 | -1.82 | -0.01 | -2.55 | -2.61 |
| $\Delta_{phe}$ leaf-soil | 0.05 | 1.68 | -5.2 | -0.88 | -1.09 |
| $\Delta_{bulk}$ leaf-soil | -6.22 | -1.62 | -6.51 | -5.3 | -4.91 |

**Table 4:** Isotope values, offsets between bulk and pheo *a* $\delta^{15}N$ values, and differences in pheo *a* and bulk $\delta^{15}N$ values across the leaf-soil continuum, across sites. Bulk values are included only for samples for which a corresponding pheo *a* measurement was made.