# Peer review of "Strong correspondence between nitrogen isotope composition of foliage and chlorin across a rainfall gradient: Implications for paleoreconstruction of the nitrogen cycle"

_Biogeosciences, 2019_

## Referee Comment (RC1) · Anonymous Referee #1 · 27 May 2019

General comments

This paper describes an initial test of the hypothesis that $\delta15N_{pheo}$ may be a good proxy for original $\delta15N_{leaf}$ values, and therefore a way to trace the history of N limitation/availability in terrestrial soils. The paper is well-written and the authors successfully demonstrate that $\delta15N_{pheo}$ is a better tracer of $\delta15N_{leaf}$ than soil $\delta15N_{bulk}$. This is an important first step toward developing such a proxy, although the authors correctly point out that additional work in a system where dating of soil horizons is possible will be necessary, in order to resolve some of the outstanding questions regarding how

$\delta$15Npheo integrates different timescales and across different plant species. The difference between plant species I found very interesting- future measurements will maybe help to define those trends in more detail. A few specific comments follow, but overall this was an easy and interesting read.

Specific comments

1. Upon initial reading of the paper I got a little confused trying to sort out exactly what $\varepsilon$pheo-bulk meant. I think now I understand it depends on the material being discussed, for example it could be either: the difference between $\delta$15Npheo of leaves and bulk $\delta$15N of leaves or the difference between $\delta$15Npheo of soil and bulk $\delta$15N of soil organic matter. But this is different from comparing the $\delta$15Npheo in soil/litter with $\delta$15Npheo and $\delta$15Nbulk of leaves. I'm not sure if anything really needs to be changed, but maybe a sentence of clarification somewhere might help?

2. The methods for HPLC should be more specific. For example what does "variable ratio" mean? (p. 6 line 24). What is the advantage of using two HPLC steps? Was the sample divided in half, then each part passed through each HPLC method/column? Or was it successive, i.e. sample goes through column A and into B?

3. I strongly suggest the authors consider depositing their data into a data repository where it can be easily accessed by anyone, in keeping with global scientific trends of making more data open access.

4. Do the authors have any theories for why trees exhibit more positive $\varepsilon$pheo-bulk of leaves compared to herbaceous plants? Cellulose accumulation is mentioned in the paper but I'm unclear how it's connected to nitrogen isotope values?

5. I read another paper recently (Wang et al 2019, GRL) that used bulk N isotope values of black carbon deposited in lake sediments as a proxy for regional N availability over the last 10,000 years. I wonder if the $\delta$15Npheo proxy in soil could be coupled or compared with that method, both to further validate both proxies but also to study

changes in N availability in more detail.

Technical corrections

1. Line 4 methods should be "factor"

2. Figure 3: is there a blue triangle where there shouldn't be in the 2500 mm precipitation category?

3. Figure 5: Color code here instead of label? Some of the labels overlap and can be hard to read.

4. Table 2: what is "Py Chl a"? I could not find a definition.

---

## Referee Comment (RC2) · Anonymous Referee #2 · 5 Jun 2019

General comments:

Goulden et al present a novel approach to understand terrestrial N dynamics by using compound specific analysis. The results show that 15Npheo is potentially a better proxy than bulk properties. The results and approaches are convincing and it definitely fits within this journal's topics. Nevertheless, as the authors also pointed out more detailed work especially in terms of the age of the material they have been working is needed to better constrain the limitations. Overall, I am impressed by the techniques used and the promising proxy presented in this study, and yet below I list some minor

[Figure]

...

...

issues I realised.

- Structurally: there are some paragraphs with only one sentence - I am not sure this is within the journal template, I recommend the authors to structure the manuscript with more concise paragraphs and better connections between paragraphs. it will be an easier read for everyone. Related to that, there are many sections in the methods and results and none in the discussion. For instance, section 2.2 and 2.3 could be combined. Accordingly, subsections in the discussion also would be better and easier to follow the flow of the discussion as in results.

- it will be probably corrected during the post-review process but still, do not forget to format the citation within the text ex: page 2, line 14 (e.g. (Drake...))

- I highly recommend authors to provide the data to databases where it is easily accessible upon publication. We should be supportive to open science and open data policies.

Introduction & discussion:

I am missing an introduction to compounds used. A nice introduction to pheophytin is only done in the discussion until I reached that point I did not really get why we are looking at pheo rather than chlorins (as the title say) and chl as it was introduced in the introduction. Overall the intro part gave a nice discussion on N dynamics in terrestrial environments, including the PNL where I was hoping to see this also in the discussion. how compound specific isotopic approaches would advance our understanding of N dynamics? what input 15Npheo will provide in terms of all the ongoing discussion? these could be implemented to discussion part in accordance with the introduction. Otherwise, the introduction could be (maybe should be) more technical and focus on more in compounds and isotopic fractionation for instance.

small details and questions:

page 3 line 13:... terrestrial d15Nleaf : leaf subscript

page 4 line 8: is climate a landscape effect? maybe precipitation is a better word?

page 4 paragraph starting from line 15 needs reconstruction, it is not an easy or maybe not well written paragraph.

page 5 line 21: first sentence is a sampling strategy should be in the below section (2.2).

Page 5 lines 29-30: (and generally many more) can authors be more specific?

page 6 line 1: what depth is the deepest soil sample from?

page 7 lines 13 and 15: JAMSTEC acronym should change places. line 15 should be in line 13

page 9 line 31 d15Npheo - o is missing

page 10 line 6: ...along the soil profile do (?) not deviate ....

page 12 lines 1-2: citation needed at this sentence where pheo is introduced.

page 12 paragraph starting with line 22: Can authors provide more info on the ages presented here? where are mentioned other sites here? close by? this paragraph and information given here can be improved.

References: please double check the format some references are all in caps lock

Figures: 1: would it be possible to indicate the vegetation somehow on these maps?

2 & 5: y axis title is cut, missing some part of 15N

Table 1: please add any other info on the sites below the letter like elevation or precipitation.

Table 3: I think the names should be written italics
* * *

---

## Author Comment (AC1) · 5 Jul 2019

Dear Anonymous Referee #1:

We appreciate your thoughtful review of our paper. Each of your comments is apt and useful for improving our paper. Below we respond to each comment and indicate how we plan to revise the manuscript accordingly. For clarity, we have copied your comments in their entirety in blue italics, and followed each with our response in normal font directly below.

1. *Upon initial reading of the paper I got a little confused trying to sort out exactly what "pheo-bulk meant. I think now I understand it depends on the material being discussed, for example it could be either: the difference between _15Npheo of leaves and bulk _15N of leaves or the difference between _15Npheo of soil and bulk _15N of soil organic matter. But this is different from comparing the _15Npheo in soil/litter with _15Npheo and _15Nbulk of leaves. I'm not sure if anything really needs to be changed, but maybe a sentence of clarification somewhere might help?*

We appreciate having this confusion pointed out and will expand the isotopic notation subsection of the methods section to provide clarifying explanation. We only use "pheo-bulk" in the paper as a subscript for $^{15}\varepsilon_{pheo\text{-}bulk}$, which we use to describe biosynthetic isotopic fractionation within whole leaf (bulk) tissue. It would therefore be more descriptive and perhaps reduce confusion to change this terms to $^{15}\varepsilon_{pheo\text{-}leaf}$ and $^{15}\varepsilon_{chl\text{-}bulk}$ to $^{15}\varepsilon_{chl\text{-}leaf}$. We use the symbol $\varepsilon$ to provide a direct comparison with previously reported values of isotopic fractionation between chlorophyll and whole leaves (Chikaraishi et al., 2005). We do discuss isotope differences between isolated compounds and bulk samples for other materials, but not using the epsilon notation. In those cases, we describe a difference in isotope values by simple subtraction, which is abbreviated with delta notation in Table 3.

2. *The methods for HPLC should be more specific. For example what does "variable ratio" mean? (p. 6 line 24). What is the advantage of using two HPLC steps? Was the sample divided in half, then each part passed through each HPLC method/column? Or was it successive, i.e. sample goes through column A and into B?*

We will expand the methods section for HPLC to provide clarity on these points. Variable ratio means that over the course of the run, the ratio of solvents flowing through the HPLC column changes. We will provide the details of how the percentages of solvents were varied throughout the run. We will clarify that sample fractions corresponding to peaks collected from the first HPLC column run were subsequently run through a second column.

3. *I strongly suggest the authors consider depositing their data into a data repository where it can be easily accessed by anyone, in keeping with global scientific trends of making more data open access.*

We agree to deposit our data into a data repository for open access by anyone. We will make our database publicly available by uploading to PANGAEA (https://www.pangaea.de/).

4. *Do the authors have any theories for why trees exhibit more positive "pheo-bulk of leaves compared to herbaceous plants? Cellulose accumulation is mentioned in the paper but I'm unclear how it's connected to nitrogen isotope values?*

We agree that our findings invite further discussion of why trees in our study exhibit greater intra-leaf isotopic fractionation between pheophytin and whole leaf tissue ($^{15}\varepsilon_{pheo-bulk}$) than plants. Differences in rates of growth or pathways of N compound synthesis, breakdown, and redistribution are processes that would have potential for N isotope fractionation. Due to their size and longevity, trees have different N storage and redistribution requirements than herbaceous plants. Although bulk leaf $\delta^{15}N$ is not observed to change on abscission (Kolb and Evans, 2002), seasonal or within-season breakdown and redistribution of foliar N compounds could involve N isotope fractionation that results in partitioning of N isotopes within leaf compounds During the growing season, chlorophyll is continually broken down and replaced. As we mention in the paper, because the energy needed to break the Mg-N bond is substantially smaller than the covalent bond, Mg loss from Chl would theoretically be expected to have little or no N isotope fractionation. However, in perennial plants with long lives such as the trees in our study, it is possible that this process happens many times, compounding expression of isotopic fractionation from this process. It is worth noting that others have also observed intra-leaf patterns in N isotope fractionation among different plant types that are not well understood. Chikaraishi et al. (2005) noted that $^{15}\varepsilon_{chl-bulk}$ of chlorophylls from C4 plants show much greater discrimination against 15N than do C3 plants, despite biosynthesis via the same pathway. We will remove mention of cellulose accumulation; while a distinguishing character of trees, cellulose contains no nitrogen and would not directly affect $\delta^{15}N$ values.

5. *I read another paper recently (Wang et al 2019, GRL) that used bulk N isotope values of black carbon deposited in lake sediments as a proxy for regional N availability over the last 10,000 years. I wonder if the _15Npheo proxy in soil could be coupled or compared with that method, both to further validate both proxies but also to study changes in N availability in more detail.*

We appreciate the encouragement to expand our discussion of how the $\delta^{15}N_{pheo}$ proxy could be applied to investigate N dynamics. We will add this to our discussion section. We see two key opportunities for application of a $\delta^{15}N_{pheo}$ to advance understanding of N dynamics here, both alone and in combination with other proxies such as the black carbon proxy used in Wang et al. 2019, GRL.

First, the ability to track changes in foliar $^{15}N$ over time gives insight into factors affecting $\delta^{15}N$ of plants, notably the availability of nitrogen. A time series of $\delta^{15}N_{pheo}$ covering periods of change in atmospheric $pCO_2$ could be obtained from an aggrading soil with dated, buried horizons such as in permafrost and used to evaluate the PNL hypothesis (and we have a paper on this in prep).

Second, comparison of the compound-specific $\delta^{15}N_{pheo}$ value with other proxy $\delta^{15}N$ values over the same time period would provide information on processes that cause them to deviate. Common sources of $\delta^{15}N$ proxy values are subaqueous sediment deposits such as from lakes, ungulate tooth enamel, and bulk wood, black carbon, or soil. Deviations in records of $\delta^{15}N$ of pheo and tooth enamel at a single site would highlight changes in factors affecting dietary fractionation, such as animal growth rate. Aquatic signals could be distinguished from terrestrial signals by comparing $\delta^{15}N_{pheo}$ from soil with $\delta^{15}N_{chl}$ of lake

sediment. $\delta^{15}N_{pheo}$ could validate a bulk proxy record such as obtained from wood or black carbon, or highlight diagenetic limitations of the record. In the case of bulk soil, combining records of $\delta^{15}N_{pheo}$ with $\delta^{15}N_{bulk}$ would provide information both on N availability to plants and dominant pathways of loss, hydrologic or gaseous, at a site, allowing for comparison of multiple N cycle dynamics over time.

Technical corrections

*1. Line 4 methods should be "factor"*

We apologize, but we cannot find what is referred to here and need to ask for clarification of the comment.

*2. Figure 3: is there a blue triangle where there shouldn't be in the 2500 mm precipitation category?*

The blue triangle is not aberrant, but we agree that it needs explanation. We will revise the figure legend to indicate that the blue trial corresponds to a sample from soil below 20 cm.

*3. Figure 5: Color code here instead of label? Some of the labels overlap and can be hard to read.*

We agree. We will add color coding and manually move labels that are overlapping.

*4. Table 2: what is "Py Chl a"? I could not find a definition.*

We will provide this definition. Py Chl a is Pyrochlorophyll a, which is likely produced from chlorophyll *a* during laboratory processing (Teng & Chen, 1999) and for this reason is combined with chlorophyll in reporting of compound abundance.

**References**

Chikaraishi, Y., Matsumoto, K., Ogawa, N. O., Suga, H., Kitazato, H., and Ohkouchi, N.: Hydrogen, carbon and nitrogen isotopic fractionations during chlorophyll biosynthesis in C3 higher plants, Phytochemistry, 66, 911-920, 10.1016/j.phytochem.2005.03.004, 2005.

Kolb, K. J., and Evans, R. D.: Implications of leaf nitrogen recycling on the nitrogen isotope composition of deciduous plant tissues, New Phytologist, 156, 57-64, 2002.

Teng, S.S. and Chen, B.H. Formation of pyrochlorophylls and their derivatives in spinach leaves during heating. Food Chemistry, 65:3, 1999.

---

## Author Comment (AC2) · 5 Jul 2019

Dear Anonymous Referee #2:

We appreciate your thoughtful review of our paper. Below we respond to each of your comments and indicate how we plan to revise the manuscript accordingly. For clarity, we have enumerated your comments and copied them in their entirety below in blue italics; we follow each with our response directly below in normal font.

1. *- Structurally: there are some paragraphs with only one sentence - I am not sure this is within the journal template, I recommend the authors to structure the manuscript with more concise paragraphs and better connections between paragraphs. it will be an easier read for everyone. Related to that, there are many sections in the methods and results and none in the discussion. For instance, section 2.2 and 2.3 could be combined. Accordingly, subsections in the discussion also would be better and easier to follow the flow of the discussion as in results.*

We place high value on readability and appreciate your specific suggestions for improving this. We are happy to add subsections to the discussion, focusing fractionation within leaves, potential as a soil-based proxy, and possible applications of the proxy. We will combine sections 2.2 and 2.3 into a single section called "Sample collection and preparation."

We are less clear on how to implement the suggestions on conciseness and flow. The sections on isotopic notation and results are the locations in our paper where we have paragraphs that either have only one sentence or are very short, and our instinct is that these are appropriate levels of conciseness. We hope that these changes, in addition to the other clarifications and rewriting we have described, will succeed in achieving an accessible and digestible manuscript. We note that our two referees both mentioned the ease of reading the current draft, but had opposite view points. We will appreciate any further comments on this front.

2. *- it will be probably corrected during the post-review process but still, do not forget to format the citation within the text ex: page 2, line 14 (e.g. (Drake...))*

We agree and will correct this in the revised manuscript.

3. *- I highly recommend authors to provide the data to databases where it is easily accessible upon publication. We should be supportive to open science and open data policies.*

We agree to do this and plan to upload our database to PANGAEA https://www.pangaea.de/.

4. *I am missing an introduction to compounds used. A nice introduction to pheophytin is only done in the discussion until I reached that point I did not really get why we are looking at pheo rather than chlorins (as the title say) and chl as it was introduced in the introduction. Overall the intro part gave a nice discussion on N dynamics in terrestrial environments, including the PNL where I was hoping to see this also in the discussion. how compound specific isotopic approaches would advance our understanding of N dynamics? what input 15Npheo will provide in terms of all the ongoing discussion? these could be implemented to discussion part in accordance with the*

*introduction. Otherwise, the introduction could be (maybe should be) more technical and focus on more in compounds and isotopic fractionation for instance.*

We appreciate this critique of missing pieces from our introduction and discussion. We agree we should provide a better introduction to the compounds used in the introduction, and to expand our discussion of how compound-specific isotopic approaches would advance understanding of N dynamics.

In the introduction section, after introducing chlorins as degradation compounds of the chlorophyll molecule, we will name pheophytin as a key chlorin of interest due to its deriving from chlorophyll breakdown in the presence of oxygen and absence of high temperatures. We will note that pheophytin is the chlorin previously found in greater relative abundance than other chlorins in organic soils and litter (Sanger, 1971). Our study found, and did not assume from the outset, that pheo a would be present in sufficient quantities for compound-specific isotopic analysis, and so we will keep further discussion of this point in the discussion section.

In the discussion section, we will discuss how the $\delta^{15}N_{pheo}$ proxy could be applied. We see two key opportunities to advance understanding of N dynamics here. First, the ability to track changes in foliar $^{15}N$ over time gives insight into factors affecting $\delta^{15}N$ of plants, notably the availability of nitrogen. A time series of $\delta^{15}N_{pheo}$ covering periods of change in atmospheric $pCO_2$ could be obtained from an aggrading soil with dated, buried horizons such as in permafrost and used to evaluate the PNL hypothesis (and we have a paper on this in prep). Second, comparison of the compound-specific $\delta^{15}N_{pheo}$ value with other proxy $\delta^{15}N$ values over the same time period would provide information on processes that cause them to deviate. Common sources of $\delta^{15}N$ proxy values are subaqueous sediment deposits such as from lakes, ungulate tooth enamel, and bulk wood or soil. Deviations in records of $\delta^{15}N$ of pheo and tooth enamel at a single site would highlight changes in factors affecting dietary fractionation, such as animal growth rate. Aquatic signals could be distinguished from terrestrial signals by comparing $\delta^{15}N_{pheo}$ from soil with $\delta^{15}N_{chl}$ of lake sediment. $\delta^{15}N_{pheo}$ could validate a bulk proxy record such as obtained from wood or black carbon, or highlight diagenetic limitations of the record. In the case of bulk soil, combining records of $\delta^{15}N_{pheo}$ with $\delta^{15}N_{bulk}$ would provide information both on N availability to plants and dominant pathways of loss, hydrologic or gaseous, at a site, allowing for comparison of multiple N cycle dynamics over time.

*page 3 line 13:... terrestrial d15Nleaf : leaf subscript*
Agreed, thank you. We will make this change.

*page 4 line 8: is climate a landscape effect? maybe precipitation is a better word?*

As we are studying impact of position on a "climosequence" along which both precipitation and temperature vary, referring to climate effects seems more appropriate than precipitation effects. We will replace "landscape effects" with "environmental effects" to avoid confusion over whether the effects we are measuring are relevant to global climate change or only to the landscape scale (product of shading, aspect, precipitation patterns, etc.).

*page 4 paragraph starting from line 15 needs reconstruction, it is not an easy or maybe not well written paragraph.*

We will rewrite this paragraph to improve parallel structure and shorten the final sentence.

*page 5 line 21: first sentence is a sampling strategy should be in the below section (2.2).*

We agree. The first sentence will be removed and the detail that the pits were dug in open, grassy areas with minimal slope added to Section 2.2 on sample collection. The second sentence will be added to the preceding paragraph where grazing is discussed.

*Page 5 lines 29-30: (and generally many more) can authors be more specific?*
We will delete the parenthetical phrase "(and generally many more)" from this sentence.

*page 6 line 1: what depth is the deepest soil sample from?*
We will a phrase to say that the deepest pit was dug to a depth of 65 cm.

*page 7 lines 13 and 15: JAMSTEC acronym should change places. line 15 should be in line 13*
We agree and will correct this in the revised manuscript.

*page 9 line 31 d15Npheo - o is missing*
We agree and will correct this in the revised manuscript.

*page 10 line 6: ...along the soil profile do (?) not deviate ....*
We agree and will correct this in the revised manuscript.

*page 12 lines 1-2: citation needed at this sentence where pheo is introduced.*
We will provide citations for both the biosynthetic and degradation pathways of pheophytin synthesis.

*page 12 paragraph starting with line 22: Can authors provide more info on the ages presented here? where are mentioned other sites here? close by? this paragraph and information given here can be improved.*
This is a good point. We will expand this paragraph to clarify that radiocarbon dates of 4130 and 8030 yBP were taken on soil organic carbon (SOC) deep within the soil profiles at climosequence sites A and D, respectively (Chadwick et al., 2007), as located in Figure 1, making them most like site C of our studied sites. All soils along the climosequence have the Hawi volcanic flow as their parent material, which cooled around 150 ky ago, and so can be considered to be the same age, though differing climate and vegetation across the range of sites would be expected to result in different rates of organic matter production, decomposition, and preservation in soil.

*References: please double check the format some references are all in caps lock*
We agree and will correct this in the revised manuscript.

*Figures: 1: would it be possible to indicate the vegetation somehow on these maps?*
Yes: we will at least be able to indicate locations of vegetation zone transitions between the sites labelled on our map. We may also be able to indicate the shape and extent of vegetation zones. Four broad zones of vegetation have been mapped on the leeward side of Kohala Mountain: lowland dry scrubland and grassland; lowland dry and mesic forest, woodland and shrubland; and wet forest and woodland (Pratt and Gon, 1998). This reference is an atlas that we are currently trying to obtain.

*2 & 5: y axis title is cut, missing some part of 15N*

We agree and will correct this in the revised manuscript. The Y axis label will be shifted to the right to avoid shaving off the top of 15N.

*Table 1: please add any other info on the sites below the letter like elevation or precipitation.*
Agreed. Average precipitation values will be added below the site letters so that the key site differences are evident at a glance.

*Table 3: I think the names should be written italics*
We agree and will correct this in the revised manuscript.

**References**

Chadwick, O. A., Kelly, E. F., Hotchkiss, S. C., and Vitousek, P. M.: Precontact vegetation and soil nutrient status in the shadow of Kohala Volcano, Hawaii, Geomorphology, 89, 70-83, 10.1016/j.geomorph.2006.07.023, 2007.

Sanger, J. E.: Identification and Quantitative Measurement of Plant Pigments in Soil Humus Layers, Ecology, 52, 959-963, 1971.

---

## Author Response (AR1)

In this document, comments received by the author appear in blue. Line and page references made by the author refer to line and page numbers in the revised manuscript with tracked changes, which appears at the end of this Author's Response document.

**Comments from Marcel van der Meer, Associate Editor**

Dear Sara,

First of all I would like to thank both reviewers for reviewing your manuscript and you for your thorough reply. I think the reviewers made some good suggestions concerning your manuscript and I think you should definitely address these. I have the impressions that the spacing between paragraphs, although very clear, also let to some additional comments. I am not sure if this fits within the style of the journal. I have also tried to find "Line 4 methods", but have been unable to find it or any other place this comment could refer to. Perhaps we will find it in the next version. I had one very minor additional comment, I find it confusing to use enrichment (or depletion) for concentrations and amounts in an stable isotope paper (Page 2 line 9). I think if you address the reviewers comments as you have indicated your manuscript would be very suitable for publication in Biogeosciences.

Best regards,

**Marcel**

**Author response to Marcel van der Meer, Associate Editor**

Dear Marcel,

Thank you for your thoughtful comments. We have revised the manuscript according to your comments and those from two anonymous referees, as we detail in our responses to them below.

In regards to your comment on paragraph spacing, we looked but could not find guidance within the manuscript preparation instructions on paragraph spacing, so have not altered the spacing from what we previously submitted.

We did indeed find the missing "Line 4 methods" reference and changed this word to "factor" (page 5, line 16).

We deleted the reference to CO2 enrichment you mention on Page 2 Line 9 when we deleted this entire paragraph (page 2, lines 10-24).

Thank you for this opportunity to contribute to Biogeosciences, and thank you again for your help improving our paper.

Best regards,

Sara

**Comments from Anonymous Referee #1**

1. Upon initial reading of the paper I got a little confused trying to sort out exactly what "pheo-bulk meant. I think now I understand it depends on the material being discussed, for example it could be either: the difference between \_15Npheo of leaves and bulk \_15N of leaves or the difference between \_15Npheo of soil and bulk \_15N of soil organic matter. But this is different from comparing the \_15Npheo in soil/litter with \_15Npheo and \_15Nbulk of leaves. I'm not sure if anything really needs to be changed, but maybe a sentence of clarification somewhere might help?

We appreciated having this confusion pointed out and have provided clarification in the isotopic notation subsection of the methods section (Page 9 line 5). We only use "pheo-bulk" in the paper as a subscript for  ${}^{15}\varepsilon_{pheo-bulk}$ , which we use to describe biosynthetic isotopic fractionation within whole leaf (bulk) tissue. To reduce confusion, we have changed this term to  ${}^{15}\varepsilon_{pheo-leaf}$  and  ${}^{15}\varepsilon_{chl-bulk}$  to  ${}^{15}\varepsilon_{chl-leaf}$ . We use the symbol  $\varepsilon$  to provide a direct comparison with previously reported values of isotopic fractionation between chlorophyll and whole leaves (Chikaraishi et al., 2005). We do discuss isotope differences between isolated compounds and bulk samples for other materials, but not using the epsilon notation. In those cases, we describe a difference in isotope values by simple subtraction, which is abbreviated with delta notation in Table 4.

2. The methods for HPLC should be more specific. For example what does "variable ratio" mean? (p. 6 line 24). What is the advantage of using two HPLC steps? Was the sample divided in half, then each part passed through each HPLC method/column? Or was it successive, i.e. sample goes through column A and into B?

We expanded the methods section for HPLC to provide clarity on these points (Page 7, Lines 6-20). Variable ratio means that over the course of the run, the ratio of solvents flowing through the HPLC column changes. We have detailed how the percentages of solvents were varied throughout the run. We clarified that sample fractions corresponding to peaks collected from the first HPLC column run were subsequently run through a second column.

3. I strongly suggest the authors consider depositing their data into a data repository where it can be easily accessed by anyone, in keeping with global scientific trends of making more data open access.

We have submitted our data to the open access database PANGAEA (https://www.pangaea.de/, Data submission 2019-08-01T06:12:00Z), and provide this information on Page 15, Line 21.

4. Do the authors have any theories for why trees exhibit more positive "pheo-bulk of leaves compared to herbaceous plants? Cellulose accumulation is mentioned in the paper but I'm unclear how it's connected to nitrogen isotope values?

We agree that our findings invite further discussion of why trees in our study exhibit greater intra-leaf isotopic fractionation between pheophytin and whole leaf tissue ( ${}^{15}\varepsilon_{pheo-bulk}$ ) than plants. Without going into speculation at great length within the paper, we have provided a suggestion on Page 11, Line 19 that the large fractionation in trees could be a systematic effect of their "greater size, longevity, and resulting N storage and redistribution requirements relative to herbaceous plants." We removed mention of cellulose accumulation; while a distinguishing character of trees, cellulose contains no nitrogen and would not directly affect  $\delta^{15}$ N values.

We suggest this explanation because differences in rates of growth or pathways of N compound synthesis, breakdown, and redistribution are processes that would have potential for N isotope fractionation. Due to their size and longevity, trees have different N storage and redistribution requirements than herbaceous plants. Although bulk leaf  $\delta^{15}N$  is not observed to change on abscission (Kolb and Evans, 2002), seasonal or within-season breakdown and redistribution of foliar N compounds could involve N isotope fractionation that results in partitioning of N isotopes within leaf compounds During the growing season, chlorophyll is continually broken down and replaced. As we mention in the paper, because the energy needed to break the Mg-N bond is substantially smaller than the covalent bond, Mg loss from Chl would theoretically be expected to have little or no N isotope fractionation. However, in perennial plants with long lives such as the trees in our study, it is possible that this process happens many times, compounding expression of isotopic fractionation from this process. It is worth noting that others have also observed intra-leaf patterns in N isotope fractionation among different plant types that are not well understood. Chikaraishi et al. (2005) noted that  ${}^{15}\varepsilon_{chl-bulk}$  of chlorophylls from C4 plants show much greater discrimination against 15N than do C3 plants, despite biosynthesis via the same pathway. We will remove mention of cellulose accumulation; while a distinguishing character of trees, cellulose contains no nitrogen and would not directly affect  $\delta^{15}$ N values.

5. I read another paper recently (Wang et al 2019, GRL) that used bulk N isotope values of black carbon deposited in lake sediments as a proxy for regional N availability over the last 10,000 years. I wonder if the \_15Npheo proxy in soil could be coupled or compared with that method, both to further validate both proxies but also to study changes in N availability in more detail.

We appreciate the encouragement to expand our discussion of how the  $\delta^{15}N_{pheo}$  proxy could be applied to investigate N dynamics. We have added a section to our discussion, "4.3 Implications of the  $\delta^{15}N_{pheo}$ proxy" (page 14, line 28). In it, we suggest two key opportunities for application of a soil-based  $\delta^{15}N_{pheo}$ proxy data to advance understanding of past terrestrial N dynamics. First, the ability to track changes in foliar  $\delta^{15}N$  over time implies comparatively direct insight into factors affecting  $\delta^{15}N$  of plants, notably the availability of nitrogen. Central questions concerning the timescale of N cycle responses to elevated  $CO_2$  concentration in the atmosphere, and whether availability of this limiting nutrient increases or decreases with climate change, could be explored by selecting a time series that covers changes in atmospheric pCO2 (Goulden, 2016).

Second, comparison of compound-specific  $\delta^{15}N_{pheo}$  with other  $\delta^{15}N$  proxies over the same time domain could provide insights into processes that cause them to deviate from one onother. Alternative sources of  $\delta^{15}N$  proxies include subaqueous sediment deposits in lakes, ungulate tooth enamel, and bulk wood,

black carbon, or soil. Deviations in records of  $\delta^{15}N$  of pheo and tooth enamel at a common site would highlight changes in factors affecting dietary isotope fractionations, such as animal growth rates.  $\delta^{15}N_{pheo}$  records could provide information on terrestrial sources of  $\delta^{15}N$  relevant to aquatic sediment  $\delta^{15}N$  records, and allow aquatic and terrestrial signals to be distinguished.  $\delta^{15}N_{pheo}$  could validate bulk proxy records, or highlight diagenetic limitations of the record. Within soil horizons, comparing  $\delta^{15}N_{pheo}$ with  $\delta^{15}N_{bulk}$  could provide information both on N availability to plants and dominant pathways of N loss, hydrologic or gaseous, at a site, allowing for comparison of multiple N cycle dynamics over time.

**Technical corrections**

1. Line 4 methods should be "factor"

Thank you; we have changed this (Page 5, Line 16).

**2. Figure 3: is there a blue triangle where there shouldn't be in the 2500 mm precipitation category?**

We have revised the figure legend to indicate that the blue trial corresponds to a sample from soil below 20 cm (Page 26).

**3. Figure 5: Color code here instead of label? Some of the labels overlap and can be hard to read.**

We have changed the dimensions of the figure to reduce label overlapping and improve appearance, and also manually separated the remaining overlapping labels (Page 29).

**4. Table 2: what is "Py Chl a"? I could not find a definition.**

Py Chl a is Pyrochlorophyll a. We now spell out the full word in the text (page 7 line 15 and Table 2).

**References**

Chikaraishi, Y., Matsumoto, K., Ogawa, N. O., Suga, H., Kitazato, H., and Ohkouchi, N.: Hydrogen, carbon and nitrogen isotopic fractionations during chlorophyll biosynthesis in C3 higher plants, Phytochemistry, 66, 911-920, 10.1016/j.phytochem.2005.03.004, 2005.

Kolb, K. J., and Evans, R. D.: Implications of leaf nitrogen recycling on the nitrogen isotope composition of deciduous plant tissues, New Phytologist, 156, 57-64, 2002.

**Comments from Anonymous Referee #2**

1. - Structurally: there are some paragraphs with only one sentence - I am not sure this is within the journal template, I recommend the authors to structure the manuscript with more concise paragraphs and better connections between paragraphs. it will be an easier read for everyone. Related to that, there are many sections in the methods and results and none in the discussion.

For instance, section 2.2 and 2.3 could be combined. Accordingly, subsections in the discussion also would be better and easier to follow the flow of the discussion as in results.

We appreciate these suggestions for improving readability and have implemented all that were specified. We have added subsections to the discussion, focusing fractionation within leaves, potential as a soil-based proxy, and possible applications of the proxy (Page 11, Lines 8 and 23, and Page 14, Line 28). We have combined sections 2.2 and 2.3 into a single section called "Sample collection and preparation" (Page 6, line 6).

Following our revisions, sections on isotopic notation and results are the locations in our paper where we have paragraphs that either have only one sentence or are very short, and our instinct is that these are appropriate levels of conciseness. We hope that the changes we've described above, in addition to the other clarifications and rewriting we have described, some of which lengthened or combined short paragraphs (e.g. page 6 line 2) or reduced large ones, will succeed in achieving an accessible and digestible manuscript.

**2. - it will be probably corrected during the post-review process but still, do not forget to format the citation within the text ex: page 2, line 14 (e.g. (Drake...))**

We agree and have edited several in-text citations, though the specific one referenced was deleted along with the rest of the paragraph.

3. - I highly recommend authors to provide the data to databases where it is easily accessible upon publication. We should be supportive to open science and open data policies.

We agree to do this and have uploaded our database to PANGAEA https://www.pangaea.de/.

4. I am missing an introduction to compounds used. A nice introduction to pheophytin is only done in the discussion until I reached that point I did not really get why we are looking at pheo rather than chlorins (as the title say) and chl as it was introduced in the introduction. Overall the intro part gave a nice discussion on N dynamics in terrestrial environments, including the PNL where I was hoping to see this also in the discussion. how compound specific isotopic approaches would advance our understanding of N dynamics? what input 15Npheo will provide in terms of all the ongoing discussion? these could be implemented to discussion part in accordance with the introduction. Otherwise, the introduction could be (maybe should be) more technical and focus on more in compounds and isotopic fractionation for instance.

We appreciate this critique of missing pieces from our introduction and discussion. In our introduction, we agree we should provide a better introduction to the compounds used and should avoid going into detail on PNL that is not relevant for this paper. In the discussion, we agree we should expand our discussion of how compound-specific isotopic approaches would advance understanding of N dynamics.

In the introduction section, we have explained that we examined chlorin fractions for the presence of individual compounds in sufficient abundance for isotopic analysis, and that pheophytin a (pheo *a*) is a

chlorin previously found in greater relative abundance than any other in organic soils and litter (Sanger, 1971a; Gorham and Sanger, 1967), and is therefore of particular interest (page 4, line 24

In the discussion section, we will discuss how the  $\delta^{15}N_{pheo}$  proxy could be applied. We have added a section to our discussion, "4.3 Implications of the  $\delta^{15}N_{pheo}$  proxy" (page 14, line 28). In it, we suggest two key opportunities for application of a soil-based  $\delta^{15}N_{pheo}$  proxy data to advance understanding of past terrestrial N dynamics. First, the ability to track changes in foliar  $\delta^{15}N$  over time implies comparatively direct insight into factors affecting  $\delta^{15}N$  of plants, notably the availability of nitrogen. Central questions concerning the timescale of N cycle responses to elevated CO2 concentration in the atmosphere, and whether availability of this limiting nutrient increases or decreases with climate change, could be explored by selecting a time series that covers changes in atmospheric pCO2 (Goulden, 2016).

Second, comparison of compound-specific  $\delta^{15}N_{pheo}$  with other  $\delta^{15}N$  proxies over the same time domain could provide insights into processes that cause them to deviate from one onother. Alternative sources of  $\delta^{15}N$  proxies include subaqueous sediment deposits in lakes, ungulate tooth enamel, and bulk wood, black carbon, or soil. Deviations in records of  $\delta^{15}N$  of pheo and tooth enamel at a common site would highlight changes in factors affecting dietary isotope fractionations, such as animal growth rates.  $\delta^{15}N_{pheo}$  records could provide information on terrestrial sources of  $\delta^{15}N$  relevant to aquatic sediment  $\delta^{15}N$  records, and allow aquatic and terrestrial signals to be distinguished.  $\delta^{15}N_{pheo}$  could validate bulk proxy records, or highlight diagenetic limitations of the record. Within soil horizons, comparing  $\delta^{15}N_{pheo}$ with  $\delta^{15}N_{bulk}$  could provide information both on N availability to plants and dominant pathways of N loss, hydrologic or gaseous, at a site, allowing for comparison of multiple N cycle dynamics over time.

**page 3 line 13:... terrestrial d15Nleaf : leaf subscript**

Agreed, thank you, we have made this change.

**page 4 line 8: is climate a landscape effect? maybe precipitation is a better word?**

As we are studying impact of position on a "climosequence" along which both precipitation and temperature vary, referring to climate effects seems more appropriate than precipitation effects. We have replaced "landscape effects" with "environmental effects" to avoid confusion over whether the effects we are measuring are relevant to global climate change or only to the landscape scale (product of shading, aspect, precipitation patterns, etc.) (page 4, line 22).

**page 4 paragraph starting from line 15 needs reconstruction, it is not an easy or maybe not well written paragraph.**

We have rewritten this paragraph to improve parallel structure and shorten the final sentence (page 4, lines 24 through 32).

**page 5 line 21: first sentence is a sampling strategy should be in the below section (2.2).**

We agree. We have removed this sentence and moved the detail that the pits were dug in open, grassy areas with minimal slope (page 6 line 2) to Section 2.2 on sample collection and preparation. We have moved the second sentence to the preceding paragraph where grazing is discussed (page 5 line 27).

**Page 5 lines 29-30: (and generally many more) can authors be more specific?**

We have deleted the parenthetical phrase "(and generally many more)" from this sentence.

**page 6 line 1: what depth is the deepest soil sample from?**

We have added a phrase to say that the deepest pit was dug to a depth of 65 cm (page 6, line 16).

*page 7 lines 13 and 15: JAMSTEC acronym should change places. line 15 should be* in line 13 We agree and have made this correction.

page 9 line 31 d15Npheo - o is missing We agree and have made this correction.

page 10 line 6: ...along the soil profile do (?) not deviate .... We agree and have made this correction.

**page 12 lines 1-2: citation needed at this sentence where pheo is introduced.**

We have provided citations for both the biosynthetic and degradation pathways of pheophytin synthesis.

**page 12 paragraph starting with line 22: Can authors provide more info on the ages presented here? where are mentioned other sites here? close by? this paragraph and information given here can be improved.**

We have clarified that radiocarbon dates of 4130 and 8030 yBP were taken on soil organic carbon (SOC) deep within the soil profiles at climosequence sites A and D, respectively (Chadwick et al., 2007), as located in Figure 1. All soils along the climosequence have the Hawi volcanic flow as their parent material, which cooled around 150 ky ago, and so can be considered to be the same age, though differing climate and vegetation across the range of sites would be expected to result in different rates of organic matter production, decomposition, and preservation in soil. We have additionally clarified the key point here, which is that is that these radiocarbon dates indicate that our studied soils may reflect organic contributions from several thousands of years of soil development (page 13 line 30).

References: please double check the format some references are all in caps lock

We agree and have made this correction.

**Figures: 1: would it be possible to indicate the vegetation somehow on these maps?**

Yes: we have colored our map to indicate the four vegetation zones bounded by precipitation isoclines.

2 & 5: y axis title is cut, missing some part of 15N

We agree and have made this correction.

**Table 1: please add any other info on the sites below the letter like elevation or precipitation.**

We agree and have made this correction. We added precipitation values below the site letters so that the key site differences are evident at a glance.

Table 3: I think the names should be written italics We agree and have made this correction.

**3.1 Bulk isotope and C and N concentration data**

Site-averaged bulk  $\delta^{15}$ N of all soil samples decreased with increasing precipitation (and elevation) across all sites, from an 10 average of 12.4‰ at site C to 5.1‰ at site M (Table 1). Site-averaged  $\delta^{15}$ N of litter decreased between sites C and M (4.3‰ to 2.8%), but sites F and I were higher than either of these values (9.0% and 7.2%). Average foliar  $\delta^{15}$ N likewise decreased between sites C and M (4.2‰ to 0.5‰), but site F (9.1‰) was higher than C (Fig. 3). Site-averaged bulk soil 815N values for those samples on which  $\delta^{15}N_{pheo}$  was measured follow similar trends (Table 2), though on average site values are slightly lower (8.4 vs. 8.9‰), reflecting the shallower average depth of the soil samples.

15

Soil %N increased from site C (0.2%) to site M (1.2%), and %C increased across these sites from 1.9% to 17.2%. C/N increased between these sites from 11.5 to 16.3 (Table 1). Litter %C and C/N was notably higher at site M (36.3% and 25.2) than at the other sites, although litter %N was relatively flat across all sites. Vegetation %C was likewise highest at site M (43.1%), though this value was less exceptional compared with vegetation at other sites.

**20**

Vegetation had the highest C/N (average of 19.9), followed by litter (17.2) and soil (12.4). At sites C, F, and M, C/N of litter is closer to that of vegetation than to soil, while at site I C/N of litter is closer to that of soil than to vegetation.

**3.2 Chlorin compound detection**

25

Chlorins were detected in leaf, litter, and soil samples, including in soil mineral horizons up to a depth of 32 cm (Table 2). As chlorophyll a has greater absorbance at 660 nm than Pheopheo a, direct proportionality between relative HPLC peak areas and relative compound abundance in the sample should not be assumed. In plant leaves, chlorophyll a was the dominant chlorin. Chlorophyll a was also found in smaller to trace amounts in litter and some soils. In litter, pheo a was the dominant pigment, with the exception of site M, where chlorophyll a was more abundant in the litter sample. In site C, chlorophyll a was absent from litter and soil. In soils, pheophytinpheo a was the most abundant degradation product. Pheo a, targeted for isotopic

**9**

analysis due to its superlative abundance, was present in sufficient concentration for isotopic analysis above ~20 cm in soils at most sites.

**3.3 Pheophytin a N isotope data of leaves, litter, and soil**

Intra-leaf isotope offsets—Across all plant samples,  $\delta^{15}N_{pheo}$  of live foliage was significantly, linearly correlated with bulk  $\delta^{15}N_{leaf}$ -(slope = 0.9; y-intercept = -1.1; adjusted R2 = 0.8; p = 0.000002507) (Fig. 4). {}^{15}\varepsilon\_{pheo-bulkleaf} was equal to 1.4‰ (± 2.3‰) across all plant samples (Eq 1).

Of the six sampled species, all but the two trees exhibited mean  ${}^{15}\varepsilon_{pheo-buildeaf}$  values  $\leq 1.5\%$  (Table 3):  ${}^{15}\varepsilon_{pheo-buildeaf}$  for *P. pallida* was 6.5‰ and  ${}^{15}\varepsilon_{pheo-buildeaf}$  for *M. polymorpha* was 2.5‰. If the *P. pallida* sample is omitted,  ${}^{15}\varepsilon_{pheo-buildeaf}$  drops to 0.71‰ ( $\pm 1.3\%$ ).

 $^{15}\varepsilon_{pheo-bulkleaf}$  was largest at site C (4.0‰) and smallest at site F (0.12‰). For a given species, average  $^{15}\varepsilon_{pheo-bulkleaf}$  tended either to remain flat or slightly decrease into the wettest sites (Fig. 5).

15  $\delta^{15}N_{pheo}$  offsets across leaf-litter-soil— $\delta^{15}N$  of Pheo *a* in litter was on average 0.3% higher than the  $\delta^{15}N$  of pheo *a* of live foliage at a common site. Average difference between the  $\delta^{15}N$  of bulk litter and foliage were slightly higher, at ~ 2.6% at a common site (Table 4). Pheo *a*-specific soil  $\delta^{15}N$  values were on average 1.3% higher than pheo *a* litter values at a common site; bulk  $\delta^{15}N$  soil values are 2.6% higher than bulk litter. The average offset between  $\delta^{45}N_{pheo}\delta^{15}N_{pheo}$  of soil and live foliage at a site was1.1%; the average offset between bulk soil and bulk vegetation  $\delta^{15}N$  was 4.9%.

20

10

**3.4 Soil depth profiles**

Soil δ15Nbulk was, on average, higher than the δ15Nbulk of overlying litter, and there was a slight trend of increasing δ15Nbulk with increasing depth in the upper ~25 cm of soil pits (Fig. 6). δ15Npheo of soil also displayed higher values relative to overlying litter in sites F and I and in part of the profile at site M (Fig. 6). Soil δ15Nbulk values returned to slightly more negative values deep in the profiles; particularly notable at site F. At site C, δ15Npheo along the soil profile do not deviate significantly from that of the overlying litter. At site M, δ15Npheo values increased slightly with depth in the upper profile, but then decreased while δ15Nbulk steadily increased with greater depth. In sum, δ15Npheo of soil did not follow δ15Nbulk, nor did it constantly track δ15Npheo 
[revised manuscript text omitted]